# Stimulation and Neuromodulation in the Treatment of Epilepsy

**DOI:** 10.3390/brainsci8010002

**Published:** 2017-12-21

**Authors:** Timothy Marc Eastin, Miguel Angel Lopez-Gonzalez

**Affiliations:** Department of Neurosurgery, Loma Linda University, Loma Linda, CA 92354, USA; meastin@llu.edu

**Keywords:** brain stimulation, neuromodulation, epilepsy, surgery

## Abstract

Invasive brain stimulation technologies are allowing the improvement of multiple neurological diseases that were non-manageable in the past. Nowadays, this technology is widely used for movement disorders and is undergoing multiple clinical and basic science research for development of new applications. Epilepsy is one of the conditions that can benefit from these emerging technologies. The objective of this manuscript is to review literature about historical background, current principles and outcomes of available modalities of neuromodulation and deep brain stimulation in epilepsy patients.

## 1. Introduction

It is estimated that epilepsy affects more than 50 million people worldwide. The majority of those patients (60–70%) reach an adequate control based on pharmacological treatments, while a refractory group of patients can become candidates for epilepsy surgery. In some situations, surgical resection is not feasible due to expected increased morbidity for epileptogenic focus located within highly functional cortical areas. In temporal lobe epilepsy, 30–40% of patients do not improve satisfactorily after resective surgery [1]. The failure rate may increase among reported series due to multiple factors, such as location of surgical target (frontal lobe, insula), classification of failure according to studies (strict Engel IV classification, or both Engel III and IV classification).

Neuromodulation allows the possibility to treat different pathologies as reversible and non-lesional alternatives. The term “neuromodulation” is essentially electrical stimulation of the nervous system in order to modulate or modify a specific function (as in movement disorders, pain, epilepsy), and can be delivered in different ways: through stimulation over skin surface, peripheral nerve stimulation, cortical stimulation, or deep brain stimulation. The different neuromodulation systems are connected to an implantable pulse generator where different stimulation settings can be modified (Figure 1). Different locations and targets for surgically implanted neuromodulation systems have been proposed for the management of epilepsy, and this review will analyze the success rate and outcomes within the available literature.

## 2. History of Neuromodulation in Epilepsy

The initial evidence of neuromodulation dates from 15 CE (Common Era) by Scribonius; he observed that gout pain improved by accidental contact with a torpedo fish and after this discovery, electrical shock was used to treat multiple types of pain [2]. Sir Victor Horsley, in 1886, performed the first cortical stimulation for focal lesion resection in patient with epilepsy; by 1908, while he was associated with Clark, they developed the stereotaxis frame for experimental stimulation in the lab [3]. Fedor Krause, between 1893 and 1912, continued performing cortical stimulation, producing the first accurate map of human motor strip, later improved by Foerster, a neurologist trained by Dejerine and Wernicke. Foerster later applied his neurological anatomical knowledge to perform surgical procedures [4]. Wilder Penfield learned the motor mapping techniques from Foerster in Germany. He eventually went beyond with mapping of speech, hearing, vision, and memory functions [4]. After 1948, the year when Spiegel and Wycis published their experience on human stereotaxy, the surgical activity in the stereotactic field expanded dramatically, with the development of multiple other frames and techniques [5]. The experimental models used by Hassler in the 1940s allowed him to develop the thalamic atlas with the nomenclature most widely used nowadays [6]. During the 1950s and 1960s the stereotactic lesions were commonly used in epilepsy with the evidence of change in electroencephalographic recordings during stimulation of thalamus and globus pallidus [2]. In 1955, the cerebellar cortex stimulation was studied by Cooke and Snider [7], and the first trial for chronic cerebellar stimulation that suggested decreased seizure frequency was performed by Cooper in 1973 [8]. This was followed by multiple studies for deep cerebellar [9], and thalamic centromedian nucleus stimulation [10]. The anterior thalamic nucleus stimulation was analyzed with SANTE trial (Stimulation of anterior nucleus of the thalamus in epilepsy) although not obtaining Food and Drug Administration (FDA) approval [11]. Other potential central nervous system (CNS) targets such as the hippocampus [12] and subthalamic nucleus [13] were analyzed, evolving more recently to the direct cortical stimulation or responsive neurostimulation [14,15] which was FDA-approved in 2013.

Peripheral nervous system stimulation in epilepsy was initially performed by Bailey in 1938. He developed a cat model for afferent vagal stimulation that modified electroencephalogram activity [16]. The mechanism described involved the effect in locus ceruleus and nucleus solitarius, and its initial clinical application of vagal nerve stimulation was performed in 1988 [17], obtaining FDA approval in 1997. More recently, based on hypothesis of connections between locus ceruleus, nucleus solitarius and trigeminal nucleus, the idea of external trigeminal nerve stimulation through transdermal or subcutaneous electrodes to stimulate trigeminal nerve supraorbital branches was triggered. This is not FDA approved yet [18].

## 3. Central Nervous System Stimulation

### 3.1. Cerebellar Stimulation

This was initially performed by Cooper in 1973, based on the idea of inhibitory effect over efferent pathways. He reported a 50% reduction in seizure activity in 18 out of 32 patients [8]. Experimental studies favored upper medial cerebellar cortex stimulation as a potential target for generalized seizures, while deep cerebellar nuclei could potentially control limbic seizures, although other studies did not show any effect at all [19]. Van Buren et al. [9], in a case series did not show seizure improvement, further contradicted by Velasco et al. [20], but later, a systematic review showed inconsistent outcomes [21].

### 3.2. Thalamic Stimulation

Thalamic nucleus stimulation is well established for treatment of movement disorders, such as essential tremor, with FDA approval since 1997. The initial ideas of therapeutic thalamic involvement on epileptic activity dates back from stereotactic lesions for seizure control since the 1960s [22], while pioneering deep brain stimulation for epilepsy was reported by Cooper and Upton until 1985 [23]. Two specific targets have been studied, the anterior thalamic nucleus and centromedian thalamic nucleus.

#### 3.2.1. Anterior Thalamic Nucleus

The rationale for anterior thalamic nucleus (AN) stimulation is based on its role as a primary relay nucleus of the limbic system, receiving projections from mammillary bodies, cingulum, amygdala, hippocampus and orbito-frontal cortex [24,25]. Isolated case series, with seizure suppression and acceptable safety features, prompted a larger multicenter trial, the SANTE study (stimulation of the anterior nucleus of thalamus for epilepsy) [11]. A total of 110 patients in 17 US institutions were included, and showed 40% reduction in seizure frequency in the treatment group against 15% reduction in the control group during the blinded on/off evaluation. At a 12-month open label follow-up, 54% of patients showed more than 50% seizure reduction. The complications reported included paresthesias (18.2%), infection (12.7%), hemorrhage (4.5%), status epilepticus (4.5%) and death (4.5%). The deaths were attributed to sudden unexpected death in epilepsy (SUDEP), and did not occur within 30 days after surgery. It is considered that AN deep brain stimulation can benefit frontal–temporal onset epilepsies. This is currently approved in Europe and Canada, but not FDA approved.

#### 3.2.2. Centromedian Thalamic Nucleus

The Centromedian (CM) thalamic nucleus, initially described by Luys in 1895, is considered part of the intralaminar nucleus of the thalamus, and has been associated with many physiological and pathological states [26]. Wilder and Penfield first postulated that this nucleus could modulate or ameliorate seizures. They receive extensive input from mesencephalic, pontine, and medullary reticular formation, while the output is described as diffuse and non-specific. Stimulation in these areas causes the so-called cortical recruiting effect. Low-frequency stimulation causes slow-wave electroencephalogram (EEG) activity associated with somnolence, while high-frequency stimulation results in desynchronized cortical activity, arousal and even epileptiform activity if stimulation is very high [27]. Extensive research and clinical work by Velasco et al., reported an overall benefit of CM stimulation with seizure frequency reduction of generalized tonic clonic seizures by 80–100%, and 60% for complex partial seizures [28]. These results were similar to those reported by Fisher et al. [29] and Valentin et al. [30], with more than 50% reduction of generalized seizure frequency. Overall, these results derived from case series with small numbers of patients. With current information, it appears that CM nucleus stimulation can be considered as a treatment option in refractory generalized epilepsy, although stimulation is less effective in frontal lobe epilepsies. At present, this target stimulation is not FDA approved.

### 3.3. Mesiotemporal Stimulation

The role of amygdala and hippocampus in epilepsy has been long studied in experimental models, and also from clinical experience, with surgical resections and success over medically intractable epilepsy. The hypothesis of hippocampal stimulation for epilepsy is based on in vitro studies exploring the effect of electrical stimulation showing suppression of epileptiform discharges with different forms of stimulation [31]. In experimental models, low-frequency stimulation inhibits the development and progression of seizures, while high-frequency stimulation increases threshold and latency of after-discharges [32]. Current use of responsive stimulation is giving clinical information about high frequency stimulation parameters and its efficacy. This option of neuromodulation has been considered when bilateral onset of mesial temporal epilepsy is present, or when high risk of neurocognitive deficit is expected after surgical resection.

Few clinical studies with small numbers of patients were conducted, showing overall, more than 50% seizure reduction. The initial study of hippocampal stimulation in epilepsy was conducted by Velasco et al. [12] in 2000, studying ten patients with a short follow-up before temporal lobectomy. The potential mechanism of action proposed was activation of perforant pathway with inhibitory influence on epileptogenic neurons of areas cornu ammonis 1–4 (CA1–CA4). They found that in 70% of patients, seizures were abolished. Later, the same group performed a randomized controlled trial with longer follow-up in nine patients with bitemporal seizure onset, and encountered seizure reduction of >95% in five patients with normal magnetic resonance imaging (MRI) and between 50–70% seizure reduction in four patients with presence of mesial temporal sclerosis [33]. Neuropsychological testing in those patients did not show impairment after stimulation period. Another randomized controlled trial by Tellez-Zenteno et al. [1] in 2006, studied four patients with unilateral mesial temporal epilepsy and hippocampal sclerosis that had high risk for memory loss with resective surgery. The reduction in seizure frequency was 15% during “On” periods. The comparison with seizure frequency at baseline showed a median reduction of 26% when the stimulator was “On”, and an increase of 49% when the stimulator was “Off”. Given the small number of patients in all available studies with mesial temporal stimulation, there is no robust evidence for its wide application. This option can be considered in patients with high risk of memory or neuropsychological decline after resective surgery, and currently, widespread use of responsive neurostimulation will allow for gathering of more data regarding its effectiveness.

### 3.4. Subthalamic Nucleus Stimulation

Subthalamic nucleus (STN) is a well-documented target for its clinical efficacy in medically intractable Parkinson’s disease either using frame or frameless stimulation techniques (Figure 2). The effect of stimulation for epilepsy in experimental models suggested some efficacy on seizure control. The proposed mechanism is by increasing glutamate input from STN, increasing gamma-Aminobutyric acid GABA-ergic firing from substantia nigra pars reticulata (SNr), and later, causing inhibition over the dorsal midbrain antiepileptic zone [34]. The first clinical application of STN stimulation in epilepsy was performed by Benabid et al. [13] in 2002 in one patient, showing 80% reduction in seizures for more than two years following treatment. Subsequent reports favored inferior subthalamic nucleus stimulation, but unfortunately, only included a very small number of patients and lacking of solid results. STN is also considered an experimental target in epilepsy, and will require controlled randomized trials and detailed neuropsychological evaluations in order to evaluate efficacy and safety.

### 3.5. Cortical Stimulation

Chronic subdural cortical stimulation: The use of surgical electrocorticography during functional brain mapping and lesion resection trigger after discharges that can progress to full seizures. Brief pulses of stimulation initially performed by Lesser et al. [35] were able to control after discharges; the mechanism of this effect is not clearly understood. In theory, stimulation trigger alterations to GABA, calcium channels, and extracellular potassium, that can induce depolarization. Experimental and small case reports trying to optimize stimulation parameters are still under investigation to clarify its application [36,37].

Responsive stimulation: The principle of responsive neurostimulation is the first closed-loop system available with the ability of both recording and stimulating. This system involves implantation of subdural and depth electrodes connected to a skull implant. The electrodes are able to register minutes-worth of data from sliding-window of EEG recordings, sense seizure onset and to deliver electrical activity with the objective to abort and inhibit the spread of epileptogenic activity (Figure 3). The device is intended to be placed in eloquent areas, and the initial clinical experience was by Osorio et al. [15], and after further studies, it was approved by FDA in 2013. A large multicenter trial by Morrell et al. [38] evaluated efficacy with seizure reduction of 44% from baseline at one year, and 65.7% by six years. The main complication rate encountered was implant site infection in (9.4%), and hemorrhage (4.7%). Half of the patients had temporal lobe epilepsy, and among them, the majority had bitemporal onset. The ability to sense epileptic activity, quantify frequency of seizures, seizure duration activity, preceding electroencephalographic events that vary among patients, and to deliver immediate electrical stimulation, are significant advantages over other neuromodulation systems, and can open the gateway for development and improvement of the technique.

## 4. Peripheral Nervous System Stimulation

### 4.1. Vagal Nerve Stimulation (VNS)

The inhibition of motor activity by stimulation of vagal nerve afferents was reported by Schweitzer and Wright in 1937 [39]. In 1938, a cat model was developed, where afferent vagal stimulation modified electroencephalogram activity [16]. The exact mechanism of vagal nerve stimulation to control seizures is not well understood; however, the hypothesis is based on the anatomic projections of the vagus nerve to reticular activating system, central autonomic network, limbic system and noradrenergic projection system [17,40].

It is estimated that about 80% of the vagus nerve is composed of afferent fibers arising from viscera, while around 20% is composed of efferent fibers that innervate larynx and have parasympathetic control of the heart, lungs, and gastrointestinal system [41]. The main trunk of the vagus nerve is easily exposed, surgically, within the carotid sheath between and behind the internal jugular vein and common carotid artery (Figure 4). The efferent cardiac branches join the sympathetic fibers high within the neck, and are very unlikely to reach those branches during standard dissection. The right vagal nerve innervates the sinoatrial node, and for that reason, the vagal nerve stimulator is implanted on the left side. Overall, given anatomical variability, it is important to communicate with anesthesia team at the time of initial stimulation, in order to manage remote, but possible, bradycardia and asystole.

The vagal nerve stimulation is indicated for medically refractory partial onset seizures, although commonly used for generalized seizures [42]. The largest long-term study by Morris et al. [43] showed a median seizure reduction of 35% at one year and 44% at three years. Interestingly, another smaller long-term study showed 65.7% seizure reduction at six years after implantation, and 75.7% seizure reduction at 10 years [44]. It was FDA approved since 1997, and the new model has a closed-loop system that can detect a 20% or higher sudden increase in heart rate, considering that this often signals the onset of a seizure. This new model was FDA approved in 2015. The adverse events reported are hoarseness and cough in around 50% of patients, which can later be improved by habituation or modification of stimulation settings. By two years of stimulation, reported hoarseness decreased to 19.8% [43]. Serious complications are uncommon, such as vocal cord paralysis (1%) and infection (1.5%) [45]. Recent studies show that earlier use of VNS therapy is proven to offer better long-term outcomes for children [46] which favored FDA approval in 2017 for children as young as four years old.

### 4.2. Trigeminal Nerve Stimulation

In 1976, Maksimow et al. [47] reported the interruption of seizures by the trigeminal nerve stimulation. The current device available in Europe and Canada is non-invasive, and stimulates the supraorbital and infraorbital nerves with transdermal electrodes. The rationale for its use is similar to that of VNS, theoretically generating stimulation transmitted to the reticular activating system. In a Phase II trial by Soss et al. [48], the seizure control was reported to be 27.4% at six months and 34.8% at 12 months. The bilateral stimulation is favored over unilateral stimulation.

## 5. Conclusions

The combination of pharmacological resistance in epilepsy, exhaustion, or inability to perform safe surgical resective options in some severe epilepsy cases open the highway for neuromodulation in epilepsy. There is a need for coupling the historical background of neuromodulation research with recent technological and computerized advances for a better understanding of available targets. The horizon of these technologies is huge. It is imperative to perform formal and large-scale clinical trials to give solid indications of neuromodulation and deep brain stimulation techniques in epilepsy.

## Figures and Tables

**Figure 1 brainsci-08-00002-f001:**
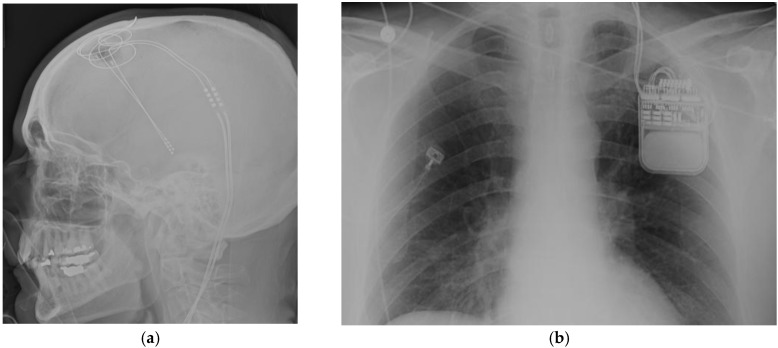
(**a**) Lateral skull X-rays view of thalamic deep brain stimulator system; (**b**) Left infraclavicular implantable pulse generator.

**Figure 2 brainsci-08-00002-f002:**
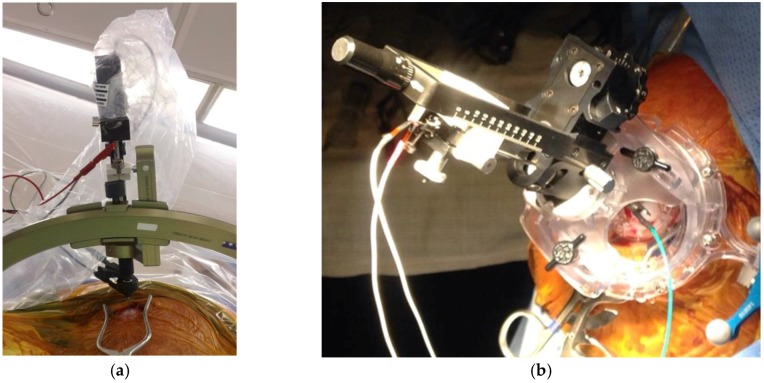
(**a**) Frame based system for deep brain stimulation; (**b**) Frameless bases system for deep brain stimulation.

**Figure 3 brainsci-08-00002-f003:**
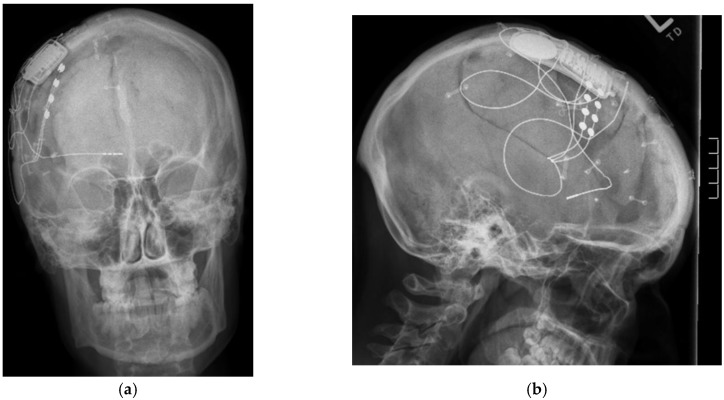
(**a**) Antero-posterior skull X-rays of implanted responsive neurostimulation; (**b**) Lateral skull X-rays of implanted responsive neurostimulation.

**Figure 4 brainsci-08-00002-f004:**
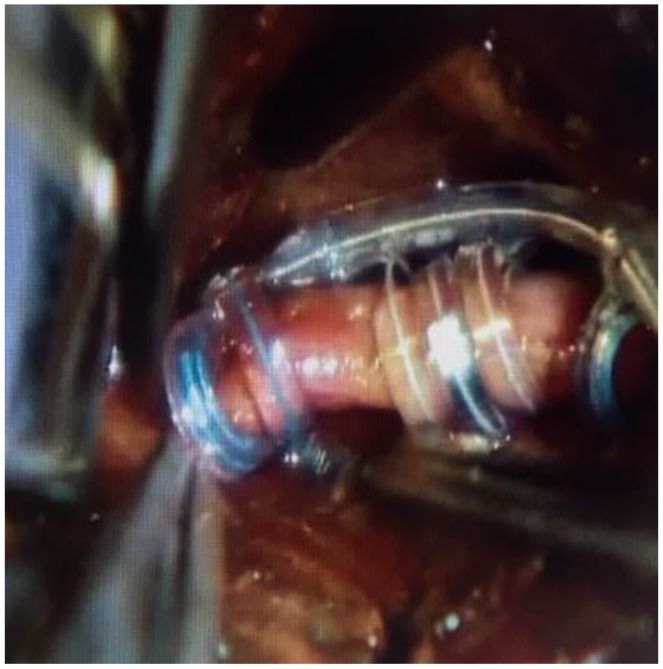
Vagus nerve stimulator placement.

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
