# Peer review of "Stimulation and Neuromodulation in the Treatment of Epilepsy"

_brainsci, 2017, doi:10.3390/brainsci8010002_

Round 1

Reviewer 1 Report

Very well written overview on modern methods of Stimulation and neuromodulation in the treatent of epilepsy. It is a critical evaluation of the modern techniques. Is is short and precise, all relevant literature is mentioned. There are a few typing mistake: page3: 3.2.2 intralaminar, 3.3. ....based on in vitro.....

Author Response

Reviewer 1 comments and suggestions (Round 1):

“Very well written overview on modern methods of Stimulation and neuromodulation in the treatent of epilepsy. It is a critical evaluation of the modern techniques. Is is short and precise, all relevant literature is mentioned. There are a few typing mistake: page3: 3.2.2 intralaminar, 3.3. ....based on in vitro.....”.

Response:

Appreciate comments by reviewer. Already corrected those mistakes suggested on page 3, section 3.2.2 Centromedian Thalamic Nucleus, and page 4, section 3.3 Mesiotemporal stimulation. Corrected also additional grammatical mistakes and punctuation evident on track changes function.

Reviewer 2 Report

The review is not comprehensive and short. Review have not discussed. There is already a paper published on similar topic. https://www.degruyter.com/downloadpdf/j/joepi.2015.23.issue-1/joepi-2015-0022/joepi-2015-0022.pdf

Author Response

Reviewer 2 comments and suggestions (Round 1):

“The review is not comprehensive and short. Review have not discussed. There is already a paper published on similar topic. https://www.degruyter.com/downloadpdf/j/joepi.2015.23.issue-1/joepi-2015-0022/joepi-2015-0022.pdf”.

Response:

Appreciate comments by reviewer. We are aware of multiple published review papers in this topic; our intention on this manuscript is to summarize only the available surgical options for stimulation and neuromodulation in epilepsy as described on page 1, section 1 Introduction. The moderate English language and style changes were made already, and are evident on track changes function. The specific changes recommended by other reviewers were made and overall improved the manuscript (specified answering reviewer 1 and 4).

Reviewer 3 Report

This is an excellent review. Well donje

Author Response

Reviewer 3 comments and suggestions (Round 1):

“This is an excellent review. Well donje”.

Response:

Appreciate comments by reviewer.

Reviewer 4 Report

Comments to authors:

Overall, this will be a useful brief review of neuromodulation as it relates to epilepsy, and I think it is very promising. In the abstract DBS is mentioned, as well as other places. At this point, perhaps framing the review in terms of invasive brain stimulation would be more helpful, rather than bringing special attention to DBS. Certainly, the review covers, as it should, areas that are not DBS. There are currently no DBS techniques FDA approved for stimulation.

It may be a bit confusing to have a section on mesiotemporal stimulation, with a concluding sentence that it is experimental. RNS is very often used for MTL stimulation. Section 3.5 could be reorganized as cortical stimulation. RNS is one type (FDA-approved, closed loop), while continuous cortical stimulation (not FDA-approved, open loop) is another (see eg Lundstrom et al, Expert Rev Neurother, 2017), which is more similar to the work of Velasco et al, as already cited. Subsections could include MTL and extra-MTL.

Minor comments:

The manuscript would benefit from careful editing. There are many awkward sentences and a few grammatical errors (see below).

L13: RCTs have shown benefit, and thus can remove “potentially” from my standpoint.

L24: “30-40%” refers only to a subset of MTL seizures, right? This number varies at least between 30 and 70% depending on many factors (location, lesion, studies, etc).

L40: Sentence splice.

L52-53: Clarify sentence.

L58: Is “conditional approval” helpful to note? Please clarify. In effect, it seems like it remains not FDA approved.

L68: Improper sentence.

L82: Was thalamic stimulation performed in the 1970s? Please cite.

L97: Again, not sure how conditional approval is helpful. Please clarify.

L97-98: “until now” could be read as it was recently approved.

L115: Again, “until now” is not clear. “At present, this target…” would be one way to express this.

L122: Is there human data to suggest this? How is the principle relevant to current RNS Neuropace programming?

L133: Clarify number of patients with 95% reduction.

L141: MTL stimulation is currently performed daily with RNS Neuropace.

L164: Only minutes worth of continuous data can be recorded, right?

L170-172: Why is this a significant advantage? How effective the closed loop portion of RNS is remains an open question. One advantage is that one can record seizure frequency over long time periods e.g. from bilateral hippocampi to determine if disabling seizures come from only one side (in the case of bilateral seizures), which could help guide a future resection. Given the limited time period of intracranial monitoring, it is clear that seizure frequency cannot be accurate gauged over a few days (see e.g. Karoly et al, Brain, 2016).

L202: Is VNS Aspire limited to detecting only 20% heart rate changes?

Author Response

Reviewer 4 comments and suggestions (Round 1):

“Overall, this will be a useful brief review of neuromodulation as it relates to epilepsy, and I think it is very promising. In the abstract DBS is mentioned, as well as other places. At this point, perhaps framing the review in terms of invasive brain stimulation would be more helpful, rather than bringing special attention to DBS. Certainly, the review covers, as it should, areas that are not DBS. There are currently no DBS techniques FDA approved for stimulation.

It may be a bit confusing to have a section on mesiotemporal stimulation, with a concluding sentence that it is experimental. RNS is very often used for MTL stimulation. Section 3.5 could be reorganized as cortical stimulation. RNS is one type (FDA-approved, closed loop), while continuous cortical stimulation (not FDA-approved, open loop) is another (see eg Lundstrom et al, Expert Rev Neurother, 2017), which is more similar to the work of Velasco et al, as already cited. Subsections could include MTL and extra-MTL.

Minor comments:

The manuscript would benefit from careful editing. There are many awkward sentences and a few grammatical errors (see below).

L13: RCTs have shown benefit, and thus can remove “potentially” from my standpoint.

L24: “30-40%” refers only to a subset of MTL seizures, right? This number varies at least between 30 and 70% depending on many factors (location, lesion, studies, etc).

L40: Sentence splice.

L52-53: Clarify sentence.

L58: Is “conditional approval” helpful to note? Please clarify. In effect, it seems like it remains not FDA approved.

L68: Improper sentence.

L82: Was thalamic stimulation performed in the 1970s? Please cite.

L97: Again, not sure how conditional approval is helpful. Please clarify.

L97-98: “until now” could be read as it was recently approved.

L115: Again, “until now” is not clear. “At present, this target…” would be one way to express this.

L122: Is there human data to suggest this? How is the principle relevant to current RNS Neuropace programming?

L133: Clarify number of patients with 95% reduction.

L141: MTL stimulation is currently performed daily with RNS Neuropace.

L164: Only minutes worth of continuous data can be recorded, right?

L170-172: Why is this a significant advantage? How effective the closed loop portion of RNS is remains an open question. One advantage is that one can record seizure frequency over long time periods e.g. from bilateral hippocampi to determine if disabling seizures come from only one side (in the case of bilateral seizures), which could help guide a future resection. Given the limited time period of intracranial monitoring, it is clear that seizure frequency cannot be accurate gauged over a few days (see e.g. Karoly et al, Brain, 2016).

L202: Is VNS Aspire limited to detecting only 20% heart rate changes?”

Response:

Appreciate detailed comments by reviewer. Agree with comment of using term of invasive brain stimulation rather than specifically deep brain stimulation in the abstract section. This is modified already page 1, Abstract, and evident on track changes draft.

Agree with the modification on Page 4, section 3.3 mesio-temporal stimulation section (track changes evident) removing the confusing comment of experimental use of this technique. We also modified section 3.5 Cortical stimulation in page 5, including suggested chronic subdural cortical stimulation section.

Careful editing was made already and evident on track changes.

Response to above specific recommendations:

L13: Removed “potentially”(line 13 after accepting track changes).

L24: Agree on above comment, this was based on reference 1 about temporal lobe epilepsy.

L40: Corrected punctuation (line 43 after accepting track changes).

L52-53: Clarified sentence from:  “…the activity in the stereotactic field exploded with development of multiple other frames and techniques”, to “…the surgical activity in the stereotactic field expanded dramatically, with the development of multiple other frames and techniques.”(line 54-55 after accepting track changes ).

L58: Agree, removed comment about conditional approval. The relevance is that anterior thalamic nucleus stimulation is not FDA approved (line 64 after accepting track changes).

L68: Agree, removed: “based on known connection…”, and modified to “based on hypothesis of connections…” (line 71 after accepting track changes).

L82: Included reference 22 about thalamic lesions to control seizures since 1960’s [          Mullan S, Vailati G, Karasick J, et al. Thalamic lesions for the control of epilepsy. A study of nine cases. Arch Neurol 1967,16(3),277-285], and clarified that pioneer deep brain stimulation was in 1985 by Cooper (line 86-88 after accepting track changes).

L97: Agree, erased, not FDA approved (line 104 after accepting track changes).

L97-98: Agree, removed “until now” (as above).

L115: Agree. Modified from “This target stimulation is not FDA approved until now”, to “At present, this target stimulation is not FDA approved” (line 121 after accepting track changes).

L122: Agree. Main evidence is from experimental models included (line 127 after accepting track changes). Included comment about high frequency stimulation on responsive stimulation (line 130-132 after accepting track changes).

L133: Included the number of patients (five) with >95% seizure reduction (line 140 after accepting track changes).

L141: Addressed (line 150 and 151 after accepting track changes).

L164: Agree, addressed (lines 181 to 184   after accepting track changes).

L170-172: The ability to sense epileptic activity, quantify frequency of seizures, seizure duration activity, preceding electroencephalographic events that vary among patients, and to deliver immediately electrical stimulation, are is a significant advantages over other neuromodulation systems, and can open the gateway for development and improvement of the technique. This included already (lines 189-193 after accepting track changes).

L202: VNS detects 20% or higher heart rate changes (lines 223-225 after accepting track changes).

Reviewer 5 Report

very good review

Round 2

Reviewer 2 Report

It is much improved now with addition of some discussion.

Reviewer 4 Report

Changes are sufficient for publication.